

# DeepSNVMiner: a sequence analysis tool to detect emergent, rare mutations in subsets of cell populations

T. Daniel Andrews[1,2], Yogesh Jeelall[1,3], Dipti Talaulikar[1,4,5], Christopher C. Goodnow[1,6,7] and Matthew A. Field[1]

[1] Department of Immunology, John Curtin School of Medical Research, Australian National University, Canberra ACT, Australia

[2] National Computational Infrastructure, Canberra ACT, Australia

[3] School of Medicine and Pharmacology, University of Western Australia, Harry Perkins Institute, Perth, Australia

[4] Haematology Translational Research Unit, Haematology Unit, ACT Pathology, Canberra ACT, Australia

[5] ANU Medical School, Australian National University, Canberra ACT, Australia

[6] Immunology Division, Garvan Institute of Medical Research, Sydney NSW, Australia

[7] St Vincent's Clinical School, University of New South Wales, Darlinghurst NSW, Australia

Corresponding author
Matthew A. Field,
matt.field@anu.edu.au

## ABSTRACT

**Background.** Massively parallel sequencing technology is being used to sequence highly diverse populations of DNA such as that derived from heterogeneous cell mixtures containing both wild-type and disease-related states. At the core of such molecule tagging techniques is the tagging and identification of sequence reads derived from individual input DNA molecules, which must be first computationally disambiguated to generate read groups sharing common sequence tags, with each read group representing a single input DNA molecule. This disambiguation typically generates huge numbers of reads groups, each of which requires additional variant detection analysis steps to be run specific to each read group, thus representing a significant computational challenge. While sequencing technologies for producing these data are approaching maturity, the lack of available computational tools for analysing such heterogeneous sequence data represents an obstacle to the widespread adoption of this technology.

**Results.** Using synthetic data we successfully detect unique variants at dilution levels of 1 in a 1,000,000 molecules, and find DeeepSNVMiner obtains significantly lower false positive and false negative rates compared to popular variant callers GATK, SAMTools, FreeBayes and LoFreq, particularly as the variant concentration levels decrease. In a dilution series with genomic DNA from two cells lines, we find DeepSNVMiner identifies a known somatic variant when present at concentrations of only 1 in 1,000 molecules in the input material, the lowest concentration amongst all variant callers tested.

**Conclusions.** Here we present DeepSNVMiner; a tool to disambiguate tagged sequence groups and robustly identify sequence variants specific to subsets of starting DNA molecules that may indicate the presence of a disease. DeepSNVMiner is an automated workflow of custom sequence analysis utilities and open source tools able to differentiate somatic DNA variants from artefactual sequence variants that likely arose during DNA amplification. The workflow remains flexible such that it may be customised to variants of the data production protocol used, and supports reproducible analysis through detailed logging and reporting of results. DeepSNVMiner

## INTRODUCTION

Deep sequencing of a restricted set of gene targets in a large population of cells has rapidly become a key application of second-generation sequencing, allowing a census of variation to be conducted on an *in vivo* biological system (*Fu et al., 2011*; *Hiatt et al., 2013*; *Jabara et al., 2011*; *Kinde et al., 2011*; *Kivioja et al., 2012*; *Schmitt et al., 2012*). Applications of this technology allow polling of sequence variation in cancer subtypes (*Forshew et al., 2012*), ascertainment of minimal residual disease (*Bidard, Weigelt & Reis-Filho, 2013*), ascertainment of malignancies or antibody specificity in the immune system (*Georgiou et al., 2014*) and observation of the emergence of drug resistant virus point-mutants (*Al-Mawsawi et al., 2014*).

The central technique in molecule tagging that allows disambiguation of these deep sequence datasets is the attachment of a random unique sequence identifier (UID) to the end(s) of input DNA, either prior to or simultaneously with amplification of target sequences (Fig. 1). Hence, even though subsequent polymerase amplification of target sequences may introduce errors, mapping these sequences to their UID sequence allows easy differentiation of sequence variation that was originally present in the input DNA from variation that has been introduced during subsequent amplification steps. Recently developed methods for molecule tagging rely on digital PCR, a process where individual DNA molecules are assessed individually (*Vogelstein & Kinzler, 1999*). Several variants of this technique have now been described (*Dressman et al., 2003*; *Ottesen et al., 2006*) with the common thread being the binding of oligonucleotide to each individual input DNA molecule prior to or during amplification. This technique is not to be confused with sample barcoding or multiplexing, a process where individual samples are tagged with small oligonucleotides and pooled in a single lane for sequencing.

In comparison to traditional massively parallel sequencing, molecule tagging has an additional step where a small unique oligonucleotide is attached to each DNA molecule prior to polymerase chain reaction (PCR) amplification. While both techniques generate huge numbers of sequenced DNA molecules in parallel a potential issue with traditional sequencing is that the introduction of erroneous base calls into a single DNA molecule can result in inaccurate sequence information being amplified in subsequent PCR steps. Such issues are not necessary prohibitive for reliable variant detection when samples are relatively homogeneous however, mainly due to the relatively low base error and PCR bias rates (*Ross et al., 2013*; *Schirmer et al., 2015*), and the ability to remove candidate PCR duplicates reads using tools such as SAMTools (*Li et al., 2009* ) or SAMBLASTER (*Faust & Hall, 2014*). When the samples being sequenced are heterogeneous however, traditional

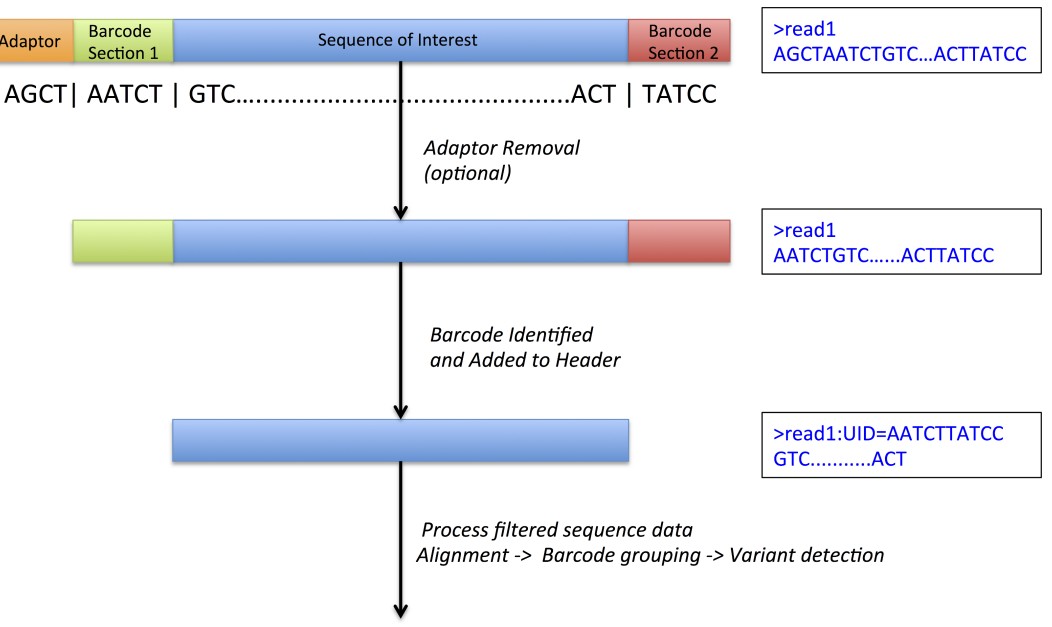

**Figure 1  DeepSNVMiner barcode and adaptor processing.** A sample sequence read which has undergone molecule tagging with a ten base pair UID consisting of five bases each attached to the 5′ and 3′ end. DeepSNVMiner first removes any adapters followed by the removal of the unique sequence identifier (UID) from the raw sequence data. The UID sequence information is preserved in the FASTQ header allowing for variant detection on read groups sharing a common UID.

variant detection methods often fail to reliably detect rare variants due to the small fraction of the original material containing the variant of interest and differences in the variant detection algorithms (*Field et al., 2015*). With molecule tagging techniques, we are able to overcome PCR issues and detect rare variants within heterogeneous samples due to the attachment of UIDs, effectively allowing the differentiation of amplification error from variation present in the original DNA molecules (*Kinde et al., 2011*).

While the utility of sequencing tagged samples is clear, the analysis of sequence data generated with UID tags is non-trivial and, as yet, software or a computational workflow does not exist in the public domain to allow easy calling and tallying of this mutation information. The fundamental technical challenge of working with such data is largely due to the wide variety of methods for attaching UIDs, methods that generate vastly different UIDs with regard to total sequence length and their position on the molecules relative to the sequence of interest and/or adaptors. The ability to work with such data requires software where users can first define the nature of the specific UID in their experiment, followed by an analysis workflow where UIDs are temporarily removed from raw sequence data for the alignment step and later restored as a means of grouping the individual reads by common UID. Finally, variants must be called within each group of input molecules sharing a common UID, a computationally intensive task given the huge numbers of groups often generated in a single experiment. To address this need we present DeepSNVMiner, a tool able to detect rare single nucleotide variants and small indels specific to a single amplified DNA molecule identified by a unique tagged sequence identifier. The DeepSNVMiner

workflow consists of grouping reads by UID tag sequences and calling variant bases in UID groups, thus identifying mutations that existed in single molecules from the original heterogeneous input DNA. DeepSNVMiner is a standalone-automated workflow that runs in a Linux or Macintosh environment and has been successfully used even on modest desktop hardware.

## MATERIALS & METHODS

### Cell lines used

Two cell lines were utilised in the dilution series experiment, HEK293 and OCI-LY10. HEK293 is available from ATCC (accession CRL-1573; http://www.atcc.org/products/all/CRL-1573.aspx) and OCI-LY10 from Ontario Cancer Institute (accession CVCL_8795; https://www.abmgood.com/OCI-Ly10-Cell-Lysate-Data-Sheet-L134.html).

### Software input

Running DeepSNVMiner requires three input files; paired-end FASTQ read files and a BED file containing the specific locations of targeted genomic region(s). An initial configuration step is also required to determine the location of three required external resources; Burrows-Wheeler Aligner (BWA) (*Li & Durbin, 2009*), SAMtools (*Li et al., 2009*), and a reference genome FASTA file with BWA index files.

### Workflow design

The workflow to disambiguate sequence variants from their unique sequence ID tags groups involves multiple steps involving both purpose built tools and calls to external binaries (Fig. 2).

First, the sequence read dataset is subjected to preliminary quality control, to remove low quality reads or those containing predominantly N calls (and hence avoid assigning UID groups of consecutive N's). The data is next interrogated for the presence of obvious adaptor sequence, which may contaminate UID tags if left untrimmed. Each UID tag is then identified based on the user defined input and removed from the FASTQ sequence line and appended to the existing FASTQ read header. These filtered reads and headers are written to new FASTQ files with the UID header information later used to detect variants specific to common UID groups. DeepSNVMiner is flexible with regard to the structure of the UID tag as both the expected UID length and strand location of the UID typically vary depending on the tagging protocol and/or sequencing technology used. For example, frequently the UID is appended solely at the 5′ end of the amplified region, but in other protocols the sequence from both the 5′ and the 3′ ends needs to be concatenated to derive the UID. Next, the modified reads are aligned to a reference genome sequence with BWA (*Li & Durbin, 2009*) using a set of alignment parameters that are permissive of mismatches but which penalise opening a gap within the alignment, especially at the ends of sequence reads. Variant bases are then identified base-by-base using the SAMTools `calmd` command (*Li et al., 2009*) within a predefined set of user-specified target genomic locations input from a BED-format file. SAMtools `calmd` is used instead of the more standard SAMtools/BCFtools workflow, as running the millions of common UID groups we typically observe through the

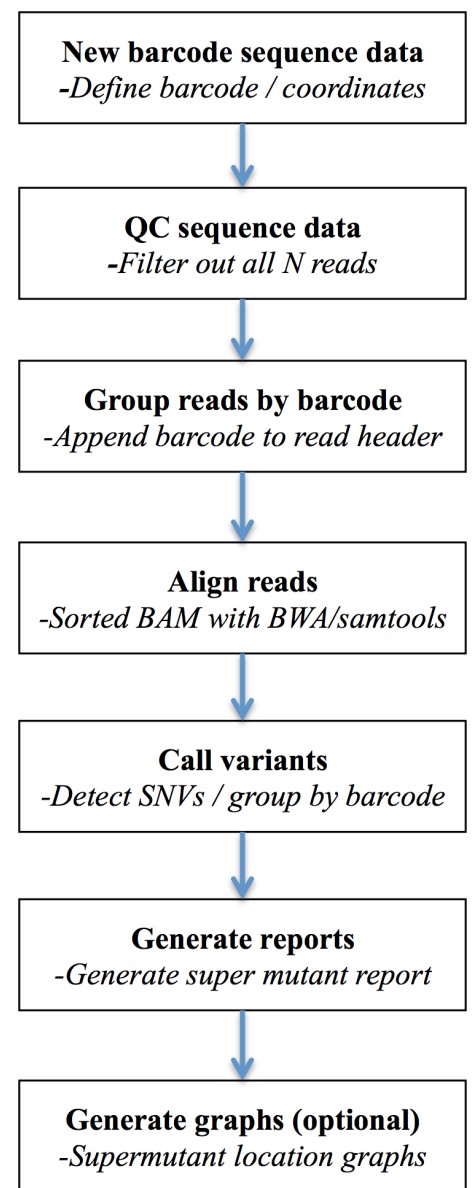

**Figure 2** **DeepSNVMiner workflow.** DeepSNVMiner Workflow consists of seven major steps: UID definition, sequence data QC, UID processing and grouping, alignment, variant detection, report generation, and optionally graphing.

entire SAMtools/BCFtools workflow is computationally prohibitive. Output from `calmd` is next parsed and variant positions and the reads in which they occur are tallied and grouped according to read UIDs. By default, common UID groups of 5 or more reads of which at least 40% detect the same variant are classified as a variant 'super mutant' and variants reported in two or more super mutants further classified as a 'super group.' This default value of 40% was chosen to allow successful super mutant detection even in the rare cases where a super mutant was missed due to an identical UID tag being added to two distinct DNA molecules. Examination of pilot data determined lowering this threshold did not add

any false positive super mutants and further, users of the software are able to override the defaults and determine the appropriate value for each cut-off based on the nature of their dataset and the method by which it was generated. Finally, optional summary graphics of the variants in the context of sequence read depth and chromosomal position are created using R.

### Workflow implementation

The overall workflow is comprised of discrete command-line interface calls to both custom- and open source-tools as well as UNIX utilities. All commands are stored within a configuration module and executed by a Perl wrapper script to allow chaining together of each command and automation of the multifaceted workflow. This also allows for easy, frequent customisation of workflow commands (should this be desired) and the capturing of specific commands and run-level information into a log file that contributes to analysis reproducibility. The workflow has the facility to allow it to be resumed or re-run from any point midway through the analysis.

The workflow commands, specifically, are various calls to several purpose-built tools (implemented in Perl), external open-source bioinformatics software tools and UNIX utilities. Custom Perl scripts are used to perform workflow steps to identify, remove and store UID tags from each read, to aggregate and summarise variant calls within UID groups and to generate final reports and graphs. Alignment of sequence reads is accomplished using BWA, variant calling is done with SAMtools `calmd`, and graphing performed with R. Identification and cleaning of reads containing runs of Ns is performed using sed and awk commands piped to other UNIX utilities such as cut, sort, uniq, and cat which are required to manipulate the output of these tools.

### Output

The final report contains a listing of all variants detected, based on either the user-configured expected variant frequency or default parameters (e.g., a common UID group must contain at least five reads with at least 40% sharing the same variant). For each called variant, the output super mutant summary reports the chromosome and genomic coordinate(s), the variant base, the UID, the number of total variant reads in the groups, the number of reads in the group and the fraction of variant reads. The super group report contains information on recurrent super mutants (grouped by common genomic coordination and variant base) and additionally reports their frequency.

## RESULTS AND DISCUSSION

We developed DeepSNVMiner to disambiguate tagged sequence groups within mixed cell populations and detect sequence variants specific to individual amplified DNA molecules. To assess the performance of DeepSNVMiner we first compare it to the well known variant callers FreeBayes, (http://arxiv.org/abs/1207.3907), Genome analysis toolkit (GATK) (*McKenna et al., 2010*), SAMTools/BCFtools (*Li, 2011*), and LoFreq (*Wilm et al., 2012*) using simulated tagged sequence data at increasing variant dilution levels. Next, we test DeepSNVMiner by running a dilution series with genomic DNA from two cells lines: one of which is known to contain a known heterozygous somatic variant.

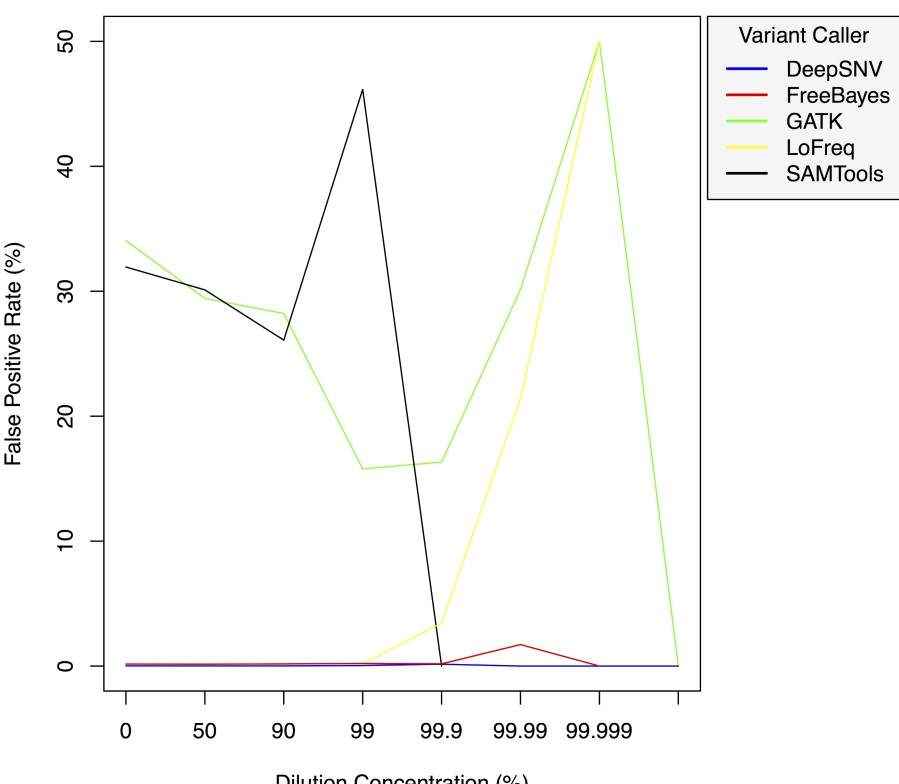

**Variant Caller False Positive Rates at Increased Dilutions**

**Figure 3** **Variant caller false positive rate at increased dilution levels.** False positive rates for Deep-SNVMiner compared to FreeBayes, GATK, LoFreq, and SAMTools at increasing variant dilution levels using synthetic data.

## Comparison with existing software

To test the effectiveness of DeepSNVMiner at increasing dilution levels two datasets containing 100-bp paired-end reads from chromosome 22 of the human reference genome (GRCh37) were created, with each read-pair having a randomly generated 10 bp barcodes attached at the 5′ end to simulate the attachment of a UID sequence tag to the original DNA molecule. The first input data set contained no mutations while the second input data set contained randomly generated single nucleotide variants (SNVs) with each mutated read duplicated randomly between 1 to 50 times within the FASTQ files to simulate the polymerase chain reaction (PCR) replication process of initial DNA fragments. Mixing the two data sets in appropriate concentrations simulated dilution levels of 0%, 50%, 90%, 99%, 99.9%, 99.99%, 99.999%, and 99.9999% with 4,000,000 million total paired end reads ultimately added to each FASTQ file. For each dilution level the FASTQ files were first aligned to chromosome 22 and variants called using DeepSNVMiner, FreeBayes, GATK, SAMTools, and LoFreq run with default parameters or as suggested in documentation (Table S1). False positive (Fig. 3) and false negative rates (Fig. 4) were then calculated (Table S2).

**Variant Caller False Negative Rates at Increased Dilutions**

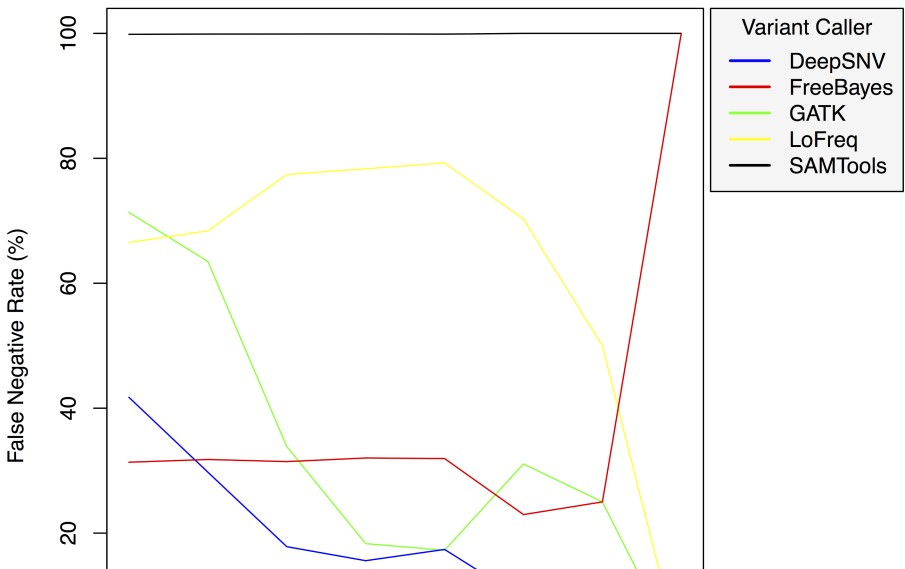

**Figure 4  Variant caller false negative rate at increased dilution levels.** False negative rates for DeepSNVMiner compared to FreeBayes, GATK, LoFreq, and SAMTools at increasing variant dilution levels using synthetic data.

**Table 1  Variant caller ability to detect known heterozygous mutation at different dilution levels.** A dilution series was performed with genomic DNA from two cells lines: HEK293 containing wild-type MYD88 and OCI-LY10 containing known heterozygous L265P MYD88 mutation. The ability to detect the known heterozygous mutation was determined for DeepSNVMiner, FreeBayes, GATK, LoFreq, and SAMTools at increasing dilution levels.

| Dilution Percent | Deep- SNVMiner | FreeBayes | GATK | LoFreq | SAMTools |
|---|---|---|---|---|---|
| 0 | Detected | Detected | Detected | Detected | Detected |
| 90 | Detected | Not detected | Detected | Detected | Not detected |
| 99 | Detected | Not detected | Not detected | Detected | Not detected |
| 99.9 | Detected | Not detected | Not detected | Not detected | Not detected |
| 99.99 | Not detected | Not detected | Not detected | Not detected | Not detected |
| 99.999 | Not detected | Not detected | Not detected | Not detected | Not detected |
| 99.9999 | Not detected | Not detected | Not detected | Not detected | Not detected |

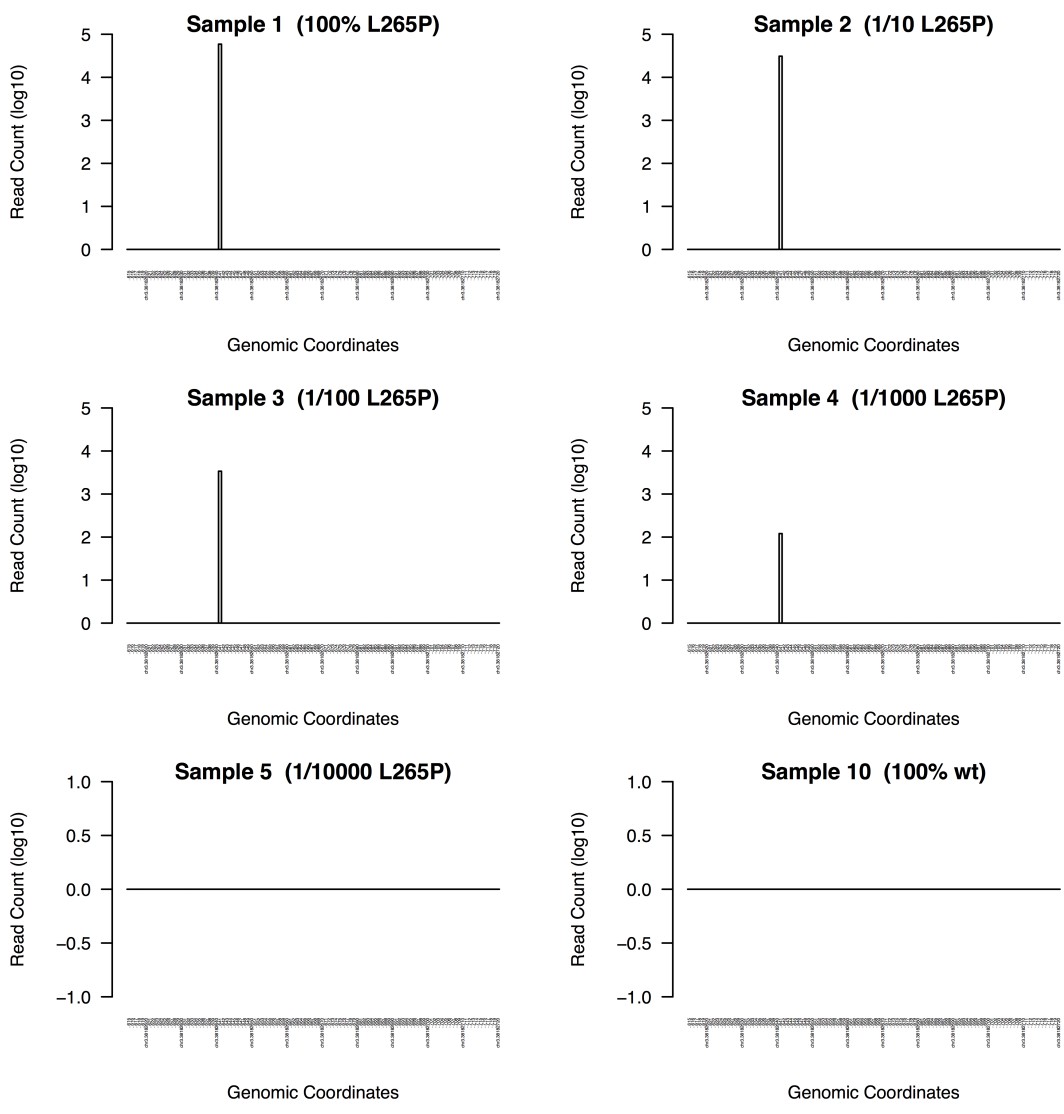

**Figure 5  DeepSNVMiner dilution series results.** To measure the sensitivity of DeepSNVMiner a dilution series containing known heterozygous L265P MYD88 mutation was performed. DeepSNVMiner was able to detect the mutation in a sample where only 1 in 1,000 samples contained the mutation.

## Real data evaluation

To evaluate the performance of DeepSNVMiner on real data versus other variant callers, we performed a dilution series using a mixture of genomic DNA from two cell lines HEK293 and OCI-LY10 (Table S3). Cell line OCI-LY10 carries a heterozygous point mutation within the *MYD88* gene at L265P or chr3:38172641 (GRCh37), a somatic mutation occurring frequently in non-Hodgkin lymphoma (*Ngo et al., 2011*). It would be clinically useful to have a method to detect and enumerate rare cells carrying this mutation in samples of blood or bone marrow. For each cell mixture in the dilution series, a 116 bp genomic region surrounding chr3:38172641 was amplified using primers with UID tags and sample ID tags and a per-sample average of 183 thousand paired-end reads were sequenced on an Illumina MiSeq. The resulting sequence reads were analysed with DeepSNVMiner,

FreeBayes, GATK, LoFreq, and SAMTools and the ability to detect the heterozygous mutation was measured.

DeepSNVMiner was successfully able to detect the mutation in dilution levels down to 1/1000 compared to 1/100 for LoFreq, 1/10 for GATK, and only in the non-diluted sample for FreeBayes and SAMTools (Table 1). In the non-diluted sample, DeepSNVMiner was able to detect the mutation in 4,055 separate super mutants consisting of 59,038 total DNA sequences and at the lower range of detection (1/1000), DeepSNVMiner detected the variant in 6 separate super mutants consisting of 120 total DNA sequences (Fig. 5)

The mutation was reliably detected at concentrations of 1/1000 by DeepSNVMiner but not in concentrations of 1/10000 indicating the lower detection limit lies somewhere in this range. However, it should be noted this limit is imposed by current laboratory methodology however, as DeepSNVMiner remains capable of achieving the theoretical limits of the technology imposed by the chosen length of UID sequences.

## CONCLUSIONS

We present DeepSNVMiner; an integrated tool set and automated workflow to allow robust and reliable identification of sequence variants present in a subset of sequences within a tagged input DNA sample. This tool makes available the analysis procedure required to support SafeSeqs and similar UID tagged sequence datasets. DeepSNVMiner has been built to allow easy automation and reproducibility and makes this technique available to a wide range of applications.

## ACKNOWLEDGEMENTS

We thank the National Computational Infrastructure (Australia) for continued access to significant computation resources and technical expertise.

### Funding

National Institutes of Health Grant U19 AI100627, NHMRC Australian Fellowship 585490, and Bioplatoforms Australia supported this work. The funders had no role in study design, data collection and analysis, decision to publish, or preparation of the manuscript.

### Grant Disclosures

The following grant information was disclosed by the authors:
National Institutes of Health Grant: U19 AI100627.
NHMRC Australian Fellowship: 585490.
Bioplatoforms Australia.

### Competing Interests

The authors declare there are no competing interests.

## Author Contributions

- T. Daniel Andrews conceived and designed the experiments, analyzed the data, wrote the paper, reviewed drafts of the paper.
- Yogesh Jeelall conceived and designed the experiments, performed the experiments, reviewed drafts of the paper.
- Dipti Talaulikar conceived and designed the experiments, contributed reagents/materials/analysis tools, reviewed drafts of the paper.
- Christopher C. Goodnow conceived and designed the experiments, reviewed drafts of the paper.
- Matthew A. Field conceived and designed the experiments, performed the experiments, analyzed the data, wrote the paper, prepared figures and/or tables, reviewed drafts of the paper.

## Data Availability

Data is available at GitHub: https://github.com/mattmattmattmatt/DeepSNVMiner

## Supplemental Information

Supplemental information for this article can be found online at http://dx.doi.org/10.7717/peerj.2074#supplemental-information.

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
