# Peer review of "DeepSNVMiner: a sequence analysis tool to detect emergent, rare mutations in subsets of cell populations"

_PeerJ, doi:10.7717/peerj.2074_

## Round 0.1 · original submission · Major Revisions

The manuscript has been carefully evaluate by two external reviewers (see attached comments). We feel that the manuscript has merit and it is providing a useful and important tool for the community. It is nicely organised and overall conforms to the journal’s standard. Nevertheless, both the reviewers raised different points that need to be carefully and properly addressed.

Reviewer 1 ·

Basic reporting

The paper by Andrews et al. describes a bioinformatics pipeline for analyzing DNA sequencing data generated from heterogeneous cell mixtures.

The manuscript is reasonably well organized and generally conforms to the journal’s standard. However, the authors should consider the following aspects to improve:
(1) In the introduction section, the authors can provide more details on how this type of deep sequencing technology works and what poses the challenge for the data analysis. One of the technical challenges is to sort out and remove the barcode and adapters. It will help if the authors provide a figure that shows how the sequence reads look like with the barcode attached and how the barcode and adapter are trimmed.
(2) Figure 2 and 3 can be combined with two sub-figures under the same scale for y-axis.
(3) For Figure 4, the authors can consider using log-transformed read counts in order to compare the results in the same scale.
(4) To make the real sequencing data used in the manuscript accessible to the public, the fastq files for the two cell lines can be deposited to Sequence Read Archive (SRA) or European Nucleotide Archive (ENA).

Experimental design

The reviewer has three questions for the pipeline design:
(1) For the preliminary quality control, how the program filters out the low quality reads, what is the threshold used in the filter?
(2) The authors use SAMTools v0.1.19 in their analysis. But the command fillmd has been replaced by calmd in latest version of SAMTools. They may need to consider updating SAMTools in their pipeline.
(3) The authors mentioned the optional summary graphics created by R as one of the functions of the pipeline, but the R code cannot be found in their github repository.
(4) Optionally, the authors can include other variant callers in their comparison, e.g. freebayes.

Validity of the findings

The authors used both simulated and real sequencing reads to test their pipeline. The major concern is why they didn’t compare the performance between variant callers in the real data.

Additional comments

No Comments.

Reviewer 2 ·

Basic reporting

The authors developed a systematic pipeline, DeepSNVMiner, to detect emergent, rare mutations in subsets of cell populations. It is always challenging to analyze population cells sequencing, considering the heterogeneity of samples. The tag-attached protocol is a potential tool to solve such problem, but the analytical solution is not well developed until DeepSNVMiner. People will take advantage of the authors works, which bridge the gap between such experimental protocol and data analysis. The manuscript is well organized and figures clearly support the content. However, due to relatively few applications of such protocol, a more detailed introduction should be very helpful to avoid the confusion for readers to understand the difference between this method and the mainstream sequencing platforms. And the data should be available for public.

Experimental design

Experimental design is logic and rigorous as well as define the original question.

Validity of the findings

Simulation data were performed very well, but the real dataset is not convincible. Is there any "real" dataset, not "diluted" one, which could be tested by the proposed workflow?

Additional comments

The fillmd is not used in the latest samtools. Authors should note this so that users can use calmd to perform analysis.

---

## Round 0.2 · Minor Revisions

The manuscript has been remarkably improved. All of the reviewers' concerns are addressed and there are just a few minor things left to adjust (see comments by reviewer 1). We would be glad to receive the revised version of the manuscript for publication in PeerJ.

Reviewer 1 ·

Basic reporting

In this revised manuscript, the authors addressed all the issues brought up by the reviewer from their original paper. The reviewer hopes they make some minor modifications on Figures 2, 3 and 4.
(1) For Figure 2 and 3, can the legend be placed outside the plotting area so that it does not block the view of the lines? If you draw the plot using R base package, you can use negative inset values for legend function.
(2) For Figure 4, it would be more striking if the y-axis of the first four bar plots has the same y limits.

Experimental design

No Comments

Validity of the findings

No Comments

Reviewer 2 ·

Basic reporting

The authors have fully addressed my previous concerns about the background of the technique, data availability and the command update. Personally, I think the draft is ready for published at Peer J.

Experimental design

No comments

Validity of the findings

No comments

---

## Round 0.3 · accepted · Accept

The manuscript is now addressing all the previous concerns and I am glad to notify the authors that this nice piece of work can be accepted for publication on PeerJ.